Genome-wide association studies for earliness, MYMIV resistance, and other associated traits in mungbean (Vigna radiata L. Wilczek) using genotyping by sequencing approach

Kohli Manju 1 2
Bansal Hina 1
Mishra Gyan Prakash gyan.gene@gmail.com 2
Dikshit Harsh Kumar harshgeneticsiari@gmail.com 2
Reddappa Shashidhar B. 2
Roy Anirban 3
Sinha Subodh Kumar 4
Shivaprasad K.M. 2
Kumari Nikki 2
Kumar Atul 5
Kumar Ranjeet R. 6
Nair Ramakrishnan Madhavan 7
Aski Muraleedhar 2
1 Amity Institute of Biotechnology, Amity University , Noida , Uttar Pradesh , India
2 Genetics, Indian Agricultural Research Institute , Delhi , Delhi , India
3 Division of Plant Pathology, Indian Agricultural Research Institute , New Delhi , Delhi , India
4 Biotechnology, National Institute of Plant Biotechnology , New Delhi , Delhi , India
5 Division of Seed Science and Technology, Indian Agricultural Research Institute , New Delhi , Delhi , India
6 Biochemistry, Indian Agricultural Research Institute , New Delhi , Delhi , India
7 World Vegetable Center, South Asia, ICRISAT Campus, Patancheru , Hyderabad , Andhra Pradesh , India
Rahman Mahmood-ur-
Electronic publication date: 2024 Jan 26
Publication date: 2024
Volume: 12
Electronic Location ID: e16653
Received 2023 Aug 2; Accepted 2023 Nov 20
Copyright: ©2024 Kohli et al.
Copyright year: 2024
Copyright holder: Kohli et al.
License: This is an open access article distributed under the terms of the Creative Commons Attribution License, which permits unrestricted use, distribution, reproduction and adaptation in any medium and for any purpose provided that it is properly attributed. For attribution, the original author(s), title, publication source (PeerJ) and either DOI or URL of the article must be cited.
License URL: https://creativecommons.org/licenses/by/4.0/

Keywords: GWAS, Mungbean, YMD, MYMIV, Flowering, Earliness

Funding: The Indian Council of Agricultural Research (ICAR), New Delhi SERB (Science and Engineering Research Board), New Delhi CRG/2019/002024 The World Vegetable Center: Taiwan, the United States Agency for International Development (USAID) The UK Government’s Foreign, Commonwealth & Development Office (FCDO) Australian Centre for International Agricultural Research (ACIAR), Germany, Thailand, Philippines, Korea, Japan ACIAR Project on International Mungbean Improvement Network CROP/2019/144 This research was funded by the Indian Council of Agricultural Research (ICAR), New Delhi, and SERB (Science and Engineering Research Board), New Delhi (CRG/2019/002024). Ramakrishnan M. Nair was supported by long-term strategic donors to the World Vegetable Center: Taiwan, the United States Agency for International Development (USAID), the UK Government’s Foreign, Commonwealth & Development Office (FCDO), the Australian Centre for International Agricultural Research (ACIAR), Germany, Thailand, Philippines, Korea, Japan, and funding from the ACIAR Project on International Mungbean Improvement Network (CROP/2019/144). The funders had no role in study design, data collection and analysis, decision to publish, or preparation of the manuscript.

==============================
Yellow mosaic disease (YMD) remains a major constraint in mungbean (Vigna radiata (L.)) production; while short-duration genotypes offer multiple crop cycles per year and help in escaping terminal heat stress, especially during summer cultivation. A comprehensive genotyping by sequencing (GBS)-based genome-wide association studies (GWAS) analysis was conducted using 132 diverse mungbean genotypes for traits like flowering time, YMD resistance, soil plant analysis development (SPAD) value, trichome density, and leaf area. The frequency distribution revealed a wide range of values for all the traits. GBS studies identified 31,953 high-quality single nucleotide polymorphism (SNPs) across all 11 mungbean chromosomes and were used for GWAS. Structure analysis revealed the presence of two genetically distinct populations based on ΔK. The linkage disequilibrium (LD) varied throughout the chromosomes and at r2 = 0.2, the mean LD decay was estimated as 39.59 kb. Two statistical models, mixed linear model (MLM) and Bayesian-information and Linkage-disequilibrium Iteratively Nested Keyway (BLINK) identified 44 shared SNPs linked with various candidate genes. Notable candidate genes identified include FPA for flowering time (VRADI10G01470; chr. 10), TIR-NBS-LRR for mungbean yellow mosaic India virus (MYMIV) resistance (VRADI09G06940; chr. 9), E3 ubiquitin-protein ligase RIE1 for SPAD value (VRADI07G28100; chr. 11), WRKY family transcription factor for leaf area (VRADI03G06560; chr. 3), and LOB domain-containing protein 21 for trichomes (VRADI06G04290; chr. 6). In-silico validation of candidate genes was done through digital gene expression analysis using Arabidopsis orthologous (compared with Vigna radiata genome). The findings provided valuable insight for marker-assisted breeding aiming for the development of YMD-resistant and early-maturing mungbean varieties.

Introduction

Mungbean (Vigna radiata (L.) Wilczek) or green gram (2n = 2x = 22; genome size: 494 to 579 Mb) is a short-season legume grown throughout the tropics and subtropics (Thomas, Fukai & Peoples, 2004; Kang et al., 2014; Liu et al., 2022). Globally, it is being grown in 7 million hectares and has approximately 5 million tons of production (Nair et al., 2019). It is known to have been cultivated in India as a native crop since prehistoric times (Vavilov, 1926); while the Central Asian region is its primary center of diversity (Kumar et al., 2014). Globally, Vigna includes nearly 100 species that are distributed in subtropical and tropical regions (Tomooka et al., 2002; Maxted et al., 2004). The legume is consumed as a cheap source of protein predominantly in the developing countries of Asia, thus helping considerably in the alleviation of protein malnutrition (Mishra et al., 2020; Nair et al., 2013). In countries such as India, Pakistan, Thailand, Indonesia, the Philippines, and China, mungbean is an integral component of cereal-based diets for a significant population (Das et al., 2020).

Yellow mosaic disease (YMD) in mungbean is transmitted by whiteflies (Bemisia tabaci Genn) and caused by begomoviruses (Mungbean Yellow Mosaic Virus (MYMV) and Mungbean Yellow Mosaic India Virus (MYMIV)). This disease causes substantial yield losses and severity may range from 10 to 100% (Marimuthu, Subramanian & Mohan, 1981; Mishra et al., 2020; Dasgupta et al., 2021). In India, YMD for the first time was reported on mungbean during 1950s at the Indian Agricultural Research Institute (IARI), New Delhi (Nariani, 1960). The MYMV is very much prevalent in Western and Southern India, Thailand, and Indonesia, while MYMIV is found in Central, Eastern, and Northern India, Bangladesh, Nepal, Pakistan, and Vietnam (Mishra et al., 2020; Dikshit et al., 2020). MYMIV is a bipartite (DNA A and DNA B), single-stranded, circular DNA (2.5–2.7 kb) virus (Nariani, 1960; Hanley-Bowdoin et al., 1999).

Various MYMIV isolates from northern India have been fully characterized by different legumes (Malathi et al., 2017). MYMIV resistance in mungbean is a complex trait, and limited efforts have been made to understand it through molecular breeding approaches. Resistance mechanisms may involve hindering viral invasion in the plant cell wall, restricting viral DNA replication, and initiating early antioxidant defense responses (Rani et al., 2017; Banerjee et al., 2018; Singh et al., 2020). Given the economic significance of mungbean and the devastating impact of YMD on its cultivation, it is crucial to intensify research efforts aimed at unraveling the molecular basis of MYMIV resistance. The relative spread of MYMIV infection using the area under disease progress curve (AUDPC) is a common method to quantify disease progression over time (Campbell & Madden, 1990).

In the wake of changing climatic conditions, the development of short-duration mungbean varieties seems a must (Badu-Apraku et al., 2017) as extra early-maturing varieties can be easily intercropped with other crops, especially cereals (Nair & Seinemachers, 2020). Leaf area and leaf number ultimately determine the light interception capacity of a crop and serve as the primary site for photosynthesis (Weraduwage et al., 2015). Thus, studies on leaf area and number of leaves concerning flowering can provide insights into the process of plant growth.

The soil-plant analysis development (SPAD) meter measures the difference between the transmittance (red-650 nm and infrared-940 nm) as a proxy of chlorophyll content (Uddling et al., 2007) which is affected by factors like cultivar, growth stage, leaf thickness, leaf color, etc. (Ata-Ul-Karim et al., 2013; Hu et al., 2014). Alam et al. (2014) reported a positive and negative relationship between SPAD and yield; and SPAD and MYMV severity, respectively. Trichomes are the “hairs” on plant stems or leaves that might be giving defense against the pest (Lemke & Mutschler, 1984; Liptay, Vandierendonck & Liptay, 1994). The trichome density may vary between genotypes and also among species (Hardin, 1979). While trichomes are generally associated with deterring or impeding whitefly infestations, there have been conflicting reports, necessitating comprehensive studies (Murugan & Nadarajan, 2012).

Genotyping is a powerful tool for the identification of variants associated with complex traits as it identifies the genetic variants as SNPs associated with the traits. The incorporation of high-throughput genotyping technologies like genotyping by sequencing (GBS) offers efficient genotyping of large populations and comprehensive coverage of the genome (Poland & Rife, 2012). GBS-based genome-wide association study (GWAS) exploits SNP markers across all chromosomes to find the linked QTLs (quantitative trait loci) with particular traits through an association mapping (AM) approach (Hirschhorn & Daly, 2005; Atwell et al., 2010).

The resistance to MYMIV in mungbean exhibits a quantitative inheritance pattern, and to date, 07 QTLs governing MYMIV resistance have been identified, spanning 04 linkage groups (LGs 2, 4, 6, 9), through bi-parental mapping (Kitsanachandee et al., 2013; Alam et al., 2014). Furthermore, Khosla et al. (2021) identified a significant QTL on chromosome 06, delineated by the Satt_281 and Sat_076 markers. Notably, Dhaliwal et al. (2022) employed the QTL-seq approach to identify a prominent QTL associated with MYMIV resistance in the blackgram and identified 03 candidate genes namely, serine-threonine kinase, UBE2D2, and BAK1/BRI1-associated receptor kinase as potential contributors to MYMIV resistance.

Globally, YMD remains a major constraint in mungbean production and its management is the biggest challenge. Due to the non-availability of diverse resistant sources to YMD, most of the commercially available cultivars were derived from only a few resistant sources (Dikshit et al., 2020; Bag et al., 2014), which has resulted in a narrow genetic base for YMD resistance. The biggest challenge in YMD management is the identification of stable QTL(s) imparting durable resistance against yellow mosaic virus (YMV) infection. Against this backdrop, this study aimed to identify the QTLs regulating an array of traits like MYMIV resistance, SPAD value, leaf area, flowering time, and trichome density in mungbean association mapping (AM) panel using GWAS approach.

Materials and Methods

Association mapping panel and phenotypic observations

The association mapping (AM) panel comprised of 132 diverse mungbean genotypes (37 released varieties, 37 advanced breeding lines, and 58 exotic germplasm lines) (Fig. S1) that were planted during kharif-2021 and kharif-2022 at Indian Agriculture Research Institute, Pusa, New Delhi (28.7041°N, 77.1025°E) in augmented design. The AM panel was derived from a larger set of nearly 300 genotypes previously studied for various traits based on the diversity of various studied traits. Standard agricultural practices were followed, except for insecticide treatments, to allow for the natural occurrence of whiteflies, as IARI is known to be a hotspot for MYMIV. The genotypes were raised on a 5.0 m plot with plant-to-plant and row-to-row spacing of 10 × 30 cm (50 plants/row) and observations were recorded for flowering time, SPAD value, leaf area, trichomes, and YMD reaction (Table 1).

Table 1 Detailed description of seven tested traits in the study.

Parameter	Description	
Days to first flowering (DFF)	First flower opening after sowing	
Days to 50% flowering (DFPF)	When plants were showing 50% flowering	
Days to 100% flowering (DHPF)	When all the plants were in full flowering condition	
The area under the disease progressive curve (AUDPC)	Disease scoring in 03 replications by calculating the infected and uninfected leaves in 03 plants	
Soil Plant Analysis Development (SPAD)	SPAD analysis was done in 03 replications (three leaf spots)	
Trichomes (TRI)	Trichomes were recorded using a compound microscope in 03 replications	
Leaf area (LA)	Leaf area was recorded in 03 replications	

To score the MYMIV infection, the AUDPC and coefficient of infection (CI) were measured at 30, 45, and 60 days after sowing (Ahmed, 1985). To maintain the sufficient disease load in the field, the infector row technique was used wherein PS16 (MYMIV susceptible) was used as a susceptible check after every 5 rows and also around the plots. AUPDC was calculated by counting the total number of leaves and also the total number of infected leaves from three infected plants of 132 genotypes constituting the AM panel (Fig. 1). CI (%) was calculated by multiplying the response value with the intensity of infection. The disease severity was calculated (30–60 days after sowing) using the following formula:

Figure 1 A representative figure showing variations in the leaf size and YMD reaction among different mungbean genotypes 45 days after sowing.

Disease severityDS={Sum of all disease ratings/ (total number of plants × maximum grade)}×100 A.U.D.P.C.= ∑in−1Yi+Yi+12xti+1−ti.

The flowering was recorded as days to first flowering (DFF), days to 50% flowering (DFPF), and days to 100% flowering (DHPF). SPAD values were recorded using SPAD 502 plus Chlorophyll Meter (Piekielek et al., 1995; Matsunaka et al., 1997) when plants were 45 days old. The trichome density was measured in the dorsal leave surface of a 30-day-old plant using a compound microscope with bright field optics (10x) and an ocular scale. The upper epidermis (middle region) of the leaf was meticulously painted with nail polish before mounting on a glass slide. Using a piece of clear sticky tape, the film was lifted from the leaf surface after five minutes. The 0–4 scale was used for the measurement of trichome density (in a unit area); where a value of 4 was marked as the highest trichome density, while 0 meant no trichome (Fig. S2). Leaf area was measured by LI-3100C-Leaf Area Meter (Engin & Engin, 2013) using 03 randomly selected leaves from a 30-day-old plant and expressed as cm2. The principal component analysis (PCA) was performed using R program.

Confirmation of causal YMD agent

To confirm the causal agent of YMD, MYMIV and MYMV-specific PCR-based amplification was performed. The MYMIV AV1 gene-specific (BM925F; 5′-AGG TGT CCC TAC CAA CAT-3′; BM926R; 5′-CCA TGG ATT GTT CCT TAC AA-3′) and MYMV gene-specific primers (AVI1F: 5′-GGA AGT GTC CCT GCC AGC G-3′and AC1R: 5′-CCA CAG GTT GAA GAA AGC AC-3′) having 497 bp and 925 bp expected amplification respectively were used on a set of susceptible mungbean genotypes. Water was used as a negative control. The PCR reaction included preheating (94 °C; 2–3 min), followed by denaturation (94 °C, 30-sec), annealing (56 °C; 45-sec), and extension (72 °C; 40-sec), repeated for 35 cycles, followed by a final extension at 72 ° C for 10 min and the product was resolved on 1.0% agarose gel.

Genotyping by sequencing assay and annotation of identified SNPs

The mungbean genotypes were grown in germination paper at room temperature and DNA was isolated from 5–6 days old seedlings (Doyle & Doyle, 1987). The sequencing was done through Illumina Hiseq 4000 using two 144-plex GBS libraries and sequenced (2 × 150 bp PE) (Bastien, Sonah & Belzile, 2014; Reddy et al., 2020). The sequencing was done through Illumina Hiseq 4000 using two 144-plex genotyping by sequencing (GBS) libraries and sequenced (2 × 150 bp PE) (Bastien, Sonah & Belzile, 2014; Reddy et al., 2020) and a comprehensive genetic coverage (150,000 reads per sample) was targeted. DNA extraction was performed using the DNeasy Plant Mini Kit, and quality of extracted DNA was confirmed through Nanodrop spectrophotometry. The library preparation was carried out using the TruSeq® DNA Nano LP kit (Illumina, San Diego, CA, USA) and was quantified using a Qubit 4.0 fluorometer with the DNA HS assay kit (ThermoFisher, Waltham, MA, USA). The sequence data is available at NCBI (PRJNA609409; Reddy et al., 2020). Then demultiplexing and mapping of FASTQ reads onto the mungbean reference genome of Kang et al. (2014) was carried out. The high-quality SNPs were detected using Bowtie v2.1.0 (Langmead & Salzberg, 2012) and reference-based GBS pipeline/genotyping approach of STACKS v1.01 (Kujur et al., 2015). The unmapped reads were analyzed using a de novo approach. The chromosomal locations of candidate genes have been identified using reference mungbean genome available at NCBI databases and data filtering using Beagle 5.4 (Browning, Zhou & Browning, 2018).

Genetic diversity, LD estimation, and marker-trait association analysis

MEGA11 (Tamura, Stecher & Kumar, 2021) was used to generate a phylogenetic tree using neighbor-joining (NJ) method and population structure was determined using STRUCTURE simulations with 20 replications at different levels of population numbers (K = 1 to 10) (STRUCTURE v2.3.4, Pritchard, Stephens & Donnelly, 2000). Tassel was used to calculate linkage disequilibrium (LD) by pooling the LD (r2) estimates and visualized through R-base. Interpretation of genome-wide SNPs was done with Q-matrix and K-matrix through general linear model (GLM) and MLM using TASSEL 5 (Saxena et al., 2014); and also using BLINK by ‘R’ program (Ihaka & Gentleman, 1996). The QQ plot was generated to determine the relative distribution of observed and expected log10(p) values for each associated SNP. Finally, Manhattan plot was generated through GWAS analysis for various studied traits like flowering time, leaf area, SPAD value, and AUDPC (Figs. S3, S4, S5).

Digital gene expression analysis for the identified candidate gene

The digital gene expression analysis was performed for the identified candidate genes using Arabidopsis orthologs. The gene expression pattern has been searched using ‘Expression Angler’, an online search tool (Austin et al., 2016) for the studied traits like flowering time, SPAD value, leaf area, MYMIV disease, etc.

Descriptive statistics

Pearson correlation coefficients were estimated using the STAR (Statistical Tool for Agricultural Research) 2.1.0 software (Gulles et al., 2014). The PCA and broad sense heritability (h2) were measured by R package (Singh & Chaudhary, 1977) using the formula Vg/Vg+Ve, where Vg and Ve are genotypic and environmental variances, respectively.

Results

Viral confirmation and phenotypic variation in the AM panel for the studied traits

The MYMIV AV1 gene-specific PCR-based amplification (497 bp) of the infected mungbean samples has confirmed the identity of the virus as MYMIV under Delhi (North Indian) conditions (Fig. S6). The frequency distribution of 132 mungbean genotypes revealed a wide range for AUDPC (1.75 to 1266.98; mean = 391.21; h2 = 99%) and CI (0.33 to 45.53; mean = 13.17; h2 = 92.55%) (Fig. 2). The onset of flowering, as measured by days to first flower (DFF) varied between 24 and 42 days (mean = 36.18 days; h2 = 68%) (Table 2). A significant positive correlation was observed between DFF and DFPF (r = 0.44); and DFF and DHPF (r = 0.3) (Fig. 3A). The DFPF ranged from 33 to 50 days (mean = 44.4 days; h2 = 80%), while DHPF ranged from 43 to 60 days (mean = 52.41 days; h2 = 82%). A negative correlation was observed between SPAD value and AUDPC (r = −0.57). Interestingly, PCA explained 47.4% of total variance from the first two PCs for the studied traits (Fig. 3B). A very broad range was recorded for the leaf area (99.42 to 221.30 cm2; mean = 146.10 cm2; h2 = 84%), SPAD value (29.33 to 62.32; mean = 46.44; h2 = 80%), and trichomes frequency (1 to 4; mean = 2.98; h2 = 46%).

Figure 2 Frequency distribution curves showing the range for (A) days to first flowering, (B) days to 50% flowering, (C) days to 100% flowering, (D) AUDPC, (E) leaf area, (F) SPAD value, and (G) trichomes.

SNP discovery

The study identified 31,953 SNPs distributed across 11 chromosomes of mungbean genome, mapping approximately 71.8% of SNPs (Reddy et al., 2020). The highest number of SNPs (4,899; 15.3%) were mapped on chromosome 1, while the lowest (1,309; 4.09%) was on chromosome 3. The SNP density varied across the chromosomes, with the lowest on chromosome 10 (0.71/100 bp; 4.71%), and the highest on chromosome 11 (9.92/100 bp; 6.15%) (Table 3).

Linkage disequilibrium and population structure analysis

LD was calculated by pooling the LD estimates (r2) using 31,953 SNPs and plotting their average r2 against 500 kb interval (maximum = 3,000 kb). LD varied throughout the chromosomes and at r2 = 0.2, the mean LD decay was 39.59 kb across all the chromosomes. With the increasing physical distance between SNPs, the LD decreased consistently (Fig. 4A). The population genetic structure of AM panel was determined using STRUCTUREv2.3.4 and all the SNPs were mapped on 11 mungbean chromosomes. The results revealed a maximum ΔK and K as 2, confirming the classification of studied genotypes into two genetically distinct populations (POP I and II) (Figs. 4B, 4C; Table S1). Identified SNPs (31,953) revealed a very high level of polymorphism and the phylogenetic tree also showed two populations (Pop I and II) as two distinct clusters (Fig. 5).

Table 2 Descriptive statistics of various studied traits among the 132 mungbean genotypes.

Trait	Mean	StDev	Maximum	Minimum	CV%	h 2 (%)	
DFF (d)	36.18	2.80	45.0	28.5	7.73	68	
DFPF (d)	44.40	3.44	52.0	32.5	7.74	80	
DHPF (d)	52.41	3.28	62.0	42.0	6.26	82	
AUDPC	391.21	311.65	1,266.98	30.14	79.66	99	
LA (cm2)	146.10	26.20	221.3	99.41	17.93	84	
SPAD	46.44	5.95	62.31	29.33	12.81	80	
TRI	2.98	0.60	4.0	1.25	20.04	46	
Notes.

StDev standard deviation

CV coefficient of variation

h2 broad sense heritability

DFF days to first flower

DFP days to 50% flowering

DHPF days to 100% flowering

AUDPC area under disease progresses curve

LA leaf area

SPAD soil plant analysis development

TRI trichomes

Figure 3 (A) Pearson correlation analysis of phenotypic traits. Where pink color indicates a complete positive correlation (r = +1), blue color indicates a complete negative correlation (r = −1), while white color indicates no correlation (r = 0). (B) Principal component analysis of 132 genotypes of mungbean for earliness and YMD incidence.

GWAS for earliness and MYMIV infection in mungbean

GWAS was conducted using 31,953 GBS-based SNPs which were integrated with phenotyping data, high-resolution population genetic structure, and PCA data. The Manhattan plot was constructed to display the results (Fig. 6). Significant marker-trait associations (MTAs) were identified using threshold [−log(p) = 3.0] as a cutoff. Both MLM and BLINK models identified 117 and 221 significant SNPs, respectively. Of these, 44 SNPs were shared by both models and were found associated with the studied traits. MLM identified 57 associated SNPs for flowering (DFF (28), DFPF (eight), and DHPF (21)) (Fig. S4); while BLINK has detected 92 SNPs associated with flowering (DFF (37), DFPF (18), and DHPF (37)). The number of significant SNPs for AUDPC, LA, SPAD, and TRI, as identified by MLM was 19, 22, 12, and seven, respectively (Table 4); while for BLINK this was 40, 52, 19, and 18, respectively (Fig. 6). The quantile–quantile plot determined the relative distribution of observed and expected log10(p) values of each SNP with the studied traits (Fig. S5).

Candidate genes for the studied traits in mungbean

To investigate the candidate genes regulating earliness (flowering time) and the yellow mosaic disease (YMD and other traits) trait in mungbean, we used 117 (MLM) and 221 (BLINK) significant SNPs. A total of 44 SNPs shared by both models were identified to be linked with 33 candidate genes having a role in the regulation of flowering time, MYMIV disease, chlorophyll formation (SPAD), leaf area, and trichomes formation in mungbean (Table S2). Primarily Arabidopsis orthologous has been identified as having a role in the regulation of earliness (flowering time) and disease resistance (Table 5).

Table 3 Details of the number of SNPs and their distribution on 11 mungbean chromosomes.

Chromosome no.	Chromosome size (bp)	SNPs per Chromosome	Average SNPs density (SNPs/10 kb)	Percent (%)	
1	3,6488,683	4,899	1.34	15.3%	
2	2,5314,989	2,751	1.08	8.60%	
3	1,2919,393	1,309	1.01	4.09%	
4	2,0774,838	2184	1.05	6.83%	
5	3,7094,413	2,871	0.77	8.98%	
6	3,7433,871	3967	1.05	12.41%	
7	5,5551,583	4,083	0.73	12.77%	
8	4,5648,752	3,797	0.83	11.88%	
9	2,1008,321	2,618	1.24	8.19%	
10	2,0995,773	1,508	0.71	4.71%	
11	1,9719,923	1,966	9.92	6.15%	
Total	33,2950,539	31,953	19.73	99.91%	

Figure 4 (A) LD decay measured in association panel of 132 diverse mungbean genotypes; (B) Δ-K plot showing the best peak at K = 2. (C) Population genetic structure plot of mungbean association mapping panel (optimal population number K = 2) with two colors.

Figure 5 Phylogenetic tree representing the genetic relationship of 132 diverse mungbean genotypes based on Nei’s genetic distance using 31,953 high-quality GBS-based SNPs.

Based on their origin, these genotypes were grouped into two main populations which are depicted in pink (Population I) and light blue color (Population II).

Figure 6 Manhattan plots showing significant p-values (BLINK model) for various traits.

Different colors were used to represent the SNPs falling on different chromosomes while horizontal lines indicate a common significance level of p = 0.0001 [−log(p) = 3.0].

A candidate gene (VRADI09G06940) regulating YMD resistance was identified as belonging to the disease resistance protein family (TIR-NBS-LRR class) on chromosome 9 (region Vr09:10949253-10955046). Also, the Senescence/dehydration-associated protein-related gene (VRADI01G03610) on chromosome 1 (Vr01:5710582-5714237) and pentatricopeptide repeat (PPR) superfamily protein gene (Vradi05g01370) on chromosome 5 (Vr05:1506441-1510662) were found associated with the imposition of disease resistance. For flowering several candidate genes have been identified, like flowering time control protein FPA gene (VRADI10G01470; chr: 10; Vr10:4265856-4270038), EARLY FLOWERING 3 gene (VRADI05G11900; chr: 5; Vr05:20836591-20841432), E3 ubiquitin-protein ligase DRIP2 gene (VRADI03G01320; chr: 3; Vr03:1791098-1797025) and homeobox-leucine zipper protein HAT22-like gene (VRADI05G11940; chr: 5; Vr05:20891041-20891644). For leaf area, GWAS has identified candidate genes like WRKY family transcription factor (VRADI03G06560; chr: 3; Vr03:7968517-7970262), and gibberellin 20 oxidase 2-like (VRADI03G06690; chr: 3; Vr03:8092219-8095631). For SPAD, E3 ubiquitin protein ligase RIE1 gene (VRADI07G28100; chr: 11; Vr11:18925287-18934324) and for trichomes, LOB domain-containing protein 21 gene (VRADI06G04290; chr: 6; Vr06:5012124-5012669) and a receptor kinase 2 gene (VRADI08G19670; chr: 8; Vr08:41687381-41723765) were identified.

Validation of identified candidate genes through digital gene expression analysis

Digital gene expression analysis helped in the in-silico validation of candidate genes using Arabidopsis orthologous (compared with the Vigna radiata genome) by identifying which part of the plant gets affected by the candidate genes. Digital gene expression analysis showed highest expression of the genes like flowering time control protein FPA (AT4G16280), EARLY FLOWERING3 (AT3G21320), homeobox-leucine zipper protein HAT22 (AT4G17460), K-transporter 1 (AT4G23640), Ca-dependent protein kinase 6 (AT4G23650), and E3 ubiquitin-protein ligase DRIP2 (AT2G30580) in the flowering tissues (Fig. 7A). Genes having the highest digital expression in leaves during MYMIV infection include disease resistance protein (TIR-NBS-LRR class) family (AT5G36930), 30S ribosomal protein S31 (AT2G21290), pentatricopeptide repeat (PPR) superfamily protein (AT1G11290) and E3 ubiquitin-protein ligase (Fig. 7B). The genes like RIE1 (AT2G01735) for SPAD (leaf senescent), WRKY family transcription factor (AT1G30650) for leaf area, LOB domain-containing protein 21 (AT3G11090) for trichomes showed affecting flowering and other plant developmental stages when studied through digital gene expression analysis (Fig. S7).

Discussion

Phenotypic variations in the studied genotypes

The results found a significant relationship between earliness and yellow mosaic disease (YMD) among 132 mungbean genotypes under field conditions. A wide AUDPC (30.15 (resistant) to 1,266.98 (susceptible)) and CI range (1.75 to 45.53%) have been recorded in the studied genotypes after 30, 45, and 60 days of sowing. These results were comparable to that of Sandhu & Dhaliwal (2019) who reported AUDPC in mungbean as 205.60 after 75 days of sowing. Similarly, Devi et al. (2017) reported AUDPC value in blackgram ranging from 105.09 (resistant) to 1,684.80 (susceptible). Bag et al. (2014) also reported a very similar range of AUDPC (347.8 to 3,000) and CI (3.6 to 69.2) in mungbean. Very high broad sense heritability for MYMIV disease (h2 = 99%) was recorded in our study and a very similar result was also reported (h2 = 99.4%) in blackgram (Thirumalai & Murugan, 2020). However, relatively low broad-sense heritability for MYMIV resistance in mungbean was reported at Gazipur (63.64%) and Madaripur (79.68%) (Alam, Somta & Srinives, 2014b). Similarly, Chen et al. (2013) observed 79% heritability for MYMIV resistance. The results suggested that the genetic variables are important in MYMIV resistance and breeding is possible using standard selection procedures.

The range of DFF (24 to 42 d), DFPF (31 to 51 d), and DHPF (43 to 63 d) was very similar to that reported by Sardar et al. (2019) in mungbean (DFF: 33 to 44 d; DFPF: 38 to 50 d; DHPF: 62 to 64 d). The broad sense heritability (h2) for DFF (68%), DFPF (80%), and DHPF (82%) was very similar to that recorded by Degefa, Petros & Andargie (2014) as DFF (87%), DFPF (97.79%) and DHPF (97.79%), respectively. Similarly, Begum et al. (2013) reported similar h2 for DFF (88.75%), DFPF (97.79%) and DHPF (88.77%). Also, Tripathi et al. (2020) reported similar h2 for DFF (62.5%), DFPF (83.6%) and DHPF (95%). Leaf area ranged for leaflets from 99.42 cm2 (33.21/leaf) to 221.30 cm2 (73.76/leaf), Similarly, Hossain et al. (2017) observed that the leaf area range concerning light in different (30, 40, and 50 days) showed range from minimum 96.29 cm2/leaflet to maximum 461.96 cm2/leaflet. Whereas Kaur & Kumar (2020) reported leaf area in mungbean ranging from 27 cm2/leaf to 30 cm2/leaf.

Table 4 Genome-wide association studies for the studied traits using a mixed linear model (MLM) and BLINK model.

Traitsa	MLM	BLINK	cTotal no. shared SNPs	
	Sig b	Average	Range	Sig	Average	Range		
	log 10 (p)	log 10 (p)		
DFF (d)	28	3.25	3.0–4.34	37	3.45	3.0–4.3	18	
DFPF (d)	8	3.18	3.0–3.5	18	3.26	3.0–3.4	0	
DHPF (d)	21	3.35	3.0–3.8	37	3.41	3.0–4.6	10	
AUDPC	19	3.49	3.0–4.4	40	3.47	3.0–4.7	11	
LA (cm2)	22	3.45	3.0–4.0	52	3.45	3.0–5.3	5	
SPAD	12	3.30	3.1–3.8	19	3.28	3.0–4.5	0	
TRI	7	3.30	3.0–3.7	18	3.18	3.0–3.7	17	
Total	117	–	–	221	–	–	44	
Notes.

DFF Days to First Flowering

DFPF Days to 50% Flowering

DHPF Days to 100% Flowering

AUDPC Area Under the Disease Progression Curve

LA Leaf Area

SPAD Soil Plant Analysis Development

TRI Trichomes

a The traits tested in the study.

b Total number of significant SNPs detected at the threshold of −log(p) = 3.0.

c The number of significant SNPs detected by both models.

Table 5 Details of candidate genes and their functions.

Vigna radiata (ID)	Chr	Associated traitsa	Start position	End position	Candidate gene details	Arabidopsis Thaliana (ID)	Function	Reference	
VRADI01G06080	1	DFF	9481001	9488781	UDP-Glycosyltransferase superfamily protein	AT2G15480	Encodes a putative glycosyltransferase which regulates flowering time via FLOWERING LOCUS C (FLC).	Holmes et al. (2012)	
VRADI02G11350	2	DFF	21338451	21339912	histone-lysine N-methyltransferase ATX3	AT1G77300	ATXR7/SDG25: Involved in flowering time regulation by activating expression of FLC.	Chen et al. (2017)	
VRADI02G11340	2	DFF	21333663	21334283	kunitz trypsin inhibitor 1	AT1G73260	It is a Kunitz trypsin inhibitor and cystatin which differentially accumulate in developing buds and floral tissues.	Pereira et al. (2011)	
VRADI02G11330	2	DFF	21310776	21327882	DNA repair and recombination protein	AT3G19210	Involved in the formation of DNA loops at the FLC locus controlling flowering time.	Jégu et al. (2011)	
VRADI03G01320	3	DFF	1791098	1797025	E3 ubiquitin protein ligase DRIP2	AT2G30580	SiGRF1 regulates the initiation of flowering by up-regulating the transcription of WRKY71 to promote Flowering Locus T (FT) and LEAFY (LFY) expression.	Wang et al. (2022)	
VRADI06G13860	6	DFF	33247347	33284542	myosin, putative	AT4G33200	Express during seedling, budding, and flowering stages.	Ahmad et al. (2022)	
VRADI10G01470	10	DFF	4265856	4270038	flowering time control protein FPA	AT4G16280	The FPA gene regulates flowering time in Arabidopsis.	Yu et al. (2021)	
VRADI07G06390	11	DFF	14606725	14608439	heat shock protein 70	AT5G28540	HSP101 regulates the expression of genes involved in flowering pathways.	Qin, Yu & Li (2021)	
VRADI08G04550	11	DFF	8575102	8578931	vacuolar cation/proton exchanger 3	AT3G51860	NHX1 and NHX2 control vacuolar pH and K+ homeostasis to regulate growth, flower development, and reproduction.	Bassil et al. (2011)	
VRADI08G04540	11	DFF	8561422	8565745	mitochondrial substrate carrier family protein B	AT3G51870	PM-ANT1: Expressed in developing pollen and mutants show impaired flower development and anther dehiscence.	Haferkamp & Schmitz-Esser (2012)	
VRADI05G11900	5	DFPF	20836591	20841432	protein EARLY FLOWERING 3	AT3G21320	EARLY FLOWERING3 encodes a novel protein regulating the circadian clock and flowering.	Hicks, Albertson & Wagner (2001)	
VRADI05G11940	5	DFPF	20891041	20891644	homeobox-leucine zipper protein HAT22 	AT4G17460	HAT or HAT4 causes early flowering, hypocotyl elongation, and altered leaf morphology.	Schena & Davis (1994)	
VRADI07G29810	7	DFPF	53528305	53535462	potassium transporter 1	AT4G23640	NaKR1 affects the accumulation of miR156 and SPL3 expression, having a role in Flowering time.	Negishi et al. (2018)	
VRADI07G29800	7	DFPF	53520228	53526097	calcium-dependent protein kinase 6	AT4G23650	CPK32 mediates Ca signaling in the regulation of flowering time in Arabidopsis.	Li et al. (2022)	
VRADI05G09940	8	DFPF	18244526	18246147	Zinc finger (Ran-binding) family protein	AT2G17975	BBX24, a Zn-finger transcription factor associated with both flowering time and stress tolerance.	Yang et al. (2014)	
VRADI01G10410	1	DHPF	20692686	20706429	U-box domain-containing protein 3	AT5G15400	The u-box gene regulates flowering time.	Sharma & Taganna (2020)	
VRADI01G10400	1	DHPF	20684812	20688192	heavy metal-associated isoprenylated plant protein 26	AT4G39700	HIPP25 with HIPP26 and HIPP27 were found in roots, leaves, flowers, and siliques of A. thaliana	Gautam (2017)	
VRADI01G03610	1	AUDPC	5710582	5714237	Senescence/dehydration-associated protein-related	AT4G15450	Senescence-associated genes induced during compatible viral interactions with grapevine and Arabidopsis.	Espinoza et al. (2007)	
VRADI02G13390	2	AUDPC	23630660	23634509	Protein phosphatase 2A regulatory B subunit family protein	AT5G25510	PP2A-B ′γ induces several defense reactions, including constitutive expression of disease resistance genes, accumulation of ROS, and premature yellowing in leaves.	Trotta et al. (2011)	
VRADI05G01370	5	AUDPC	1506441	1510662	Pentatricopeptide repeat (PPR) superfamily protein	AT1G11290	The role of PPR proteins in plant defense responses is steadily increasing.	Park et al. (2014)	
VRADI06G15850	6	AUDPC	36055395	36060459	Protein kinase superfamily protein	AT1G68690	Protein kinases catalyze reversible phosphorylation of proteins regulating plant immune response.	Jiang et al. (2022)	
VRADI09G06940	9	AUDPC	10949253	10955046	Disease resistance protein (TIR-NBS-LRR class) family	AT5G36930	NBS-LRR-encoding genes in mungbean and two wild non-progenitors reveal their role in MYMIV resistance	Purwar et al. (2023)	
VRADI09G07560	9	AUDPC	12631303	12634002	30S ribosomal protein S31	AT2G21290	Gene NbRPL12 and NbRPL19 help in delaying the response of nonhost pathogen	Nagaraj et al. (2016)	
VRADI01G03580	1	LA	5692244	5700487	DNA (cytosine-5-)-methyltransferase family
protein	AT4G08990	DNA methylation and demethylation are dynamic and strongly associated with plant development such as leaf growth, seed development, ripening, etc.	Moglia et al. (2019)	
VRADI03G06560	3	LA	7968517	7970262	WRKY family transcription factor	AT1G30650	Overexpression of WRKY15 results in an increased leaf area.	Bakshi & Oelmüller (2014)	
VRADI03G06700	3	LA	8099071	8100140	myb transcription factor	AT3G06490	MYB transcription factors are involved in cell differentiation, morphogenesis of leaves, etc.	Li et al. (2019)	
VRADI03G06690	3	LA	8092219	8095631	gibberellin 20 oxidase 2	AT1G02400	Causes longer hypocotyl, light-colored leaves, stem elongation, early flowering, etc.	Huang et al. (1998)	
VRADI02G14060	2	SPAD	24625842	24630418	Protein FAR1-RELATED SEQUENCE 4	AT4G19990	FAR1 plays multiple roles like light signal transduction, chloroplast division, chlorophyll biosynthesis, etc.	Ma & Li (2018)	
VRADI02G14090	2	SPAD	24642395	24647319	RNA-binding protein 39	AT2G18510	CP31A prefers mRNAs encoding subunits of the chloroplast NAD(P)H dehydrogenase complex.	Lenzen et al. (2020)	
VRADI04G03540	4	SPAD	7029108	7034064	proteinaceous RNase P 2	AT2G16650	miR408 gene causes morphological changes like curl leaves and sunken stomata, which could be related to decreased leaf water loss.	Hang et al. (2021)	
VRADI11G12970	11	SPAD	8925287	18934324	E3 ubiquitin protein ligase RIE1	AT2G01735	The RIE1 gene was associated with the development of leaf senescence.	Xu & Li (2003)	
VRADI06G04290	6	TRI	5012124	5012669	LOB domain-containing protein 21	AT3G11090	Associated with the development of glandular trichomes in flowering plants, including MIXTA, ATML1, and MYB106.	Chen et al. (2020)	
VRADI08G19670	7	TRI	41687381	41723765	receptor kinase 2	AT1G65800	AtRLCK VI_A3 is activated by AtROPs and is involved in trichome branching and pathogen interaction.	Reiner et al. (2015)	
Notes.

Chr Chromosome

DFPF Days to 50% Flowering

DHPF Days to 100% Flowering

AUDPC Area Under the Disease Progression Curve

LA Leaf Area

SPAD Soil Plant Analysis Development

TRI Trichomes

a Key candidate genes associated with earliness and disease-related trait.

Figure 7 Digital gene expression analysis of (A) flowering time control protein FPA, and (B) disease resistance protein (TIR-NBS-LRR class).

The h2 for leaf area was 84%, whereas Sofia et al. (2017) recorded slightly higher h2 (98.64%) for leaf area. The association between SPAD and yield was found positive, while a negative association was recorded between SPAD and MYMIV severity. Similar observations were also recorded by Alam et al. (2014). The range of SPAD value was 29.33 (highly infected) to 62.32 (low or no infection), which was comparable to Alam et al. (2014), where they have reported a range of 47 to 52 for low disease score, while 23 to 32 for high disease scoring genotypes. The h2 for SPAD was recorded as 80%, while Yimram, Somta & Srinives (2009) reported the range from 21% to 92%. Similarly, Sofia et al. (2017) reported h2 = 67.98% for SPAD in mungbean. Trichome frequency ranged from 1 to 4, and a similar trichome frequency range (3.01 to 5.31) was also reported in black gram (Taggar & Gill, 2012).

SNP discovery, linkage disequilibrium, and population structure analysis

The total length of the pseudomolecule assemblies was 332.95 Mb which accounted for 71.8% of the total mungbean genome. The SNPs mined were found dispersed in different parts of the mungbean genomes, which has resulted in varied SNP densities on the physical map of the genome. The identified SNPs (31,953) can be used for the QTL mapping for several traits, as well as comparative genome mapping involving mungbean and other legumes. The results are comparable to the number of SNPs (55,634) reported by Reddy et al. (2020) for mungbean. Furthermore, these chromosome-based SNPs can be used for marker-based mungbean breeding applications.

The identified SNPs were annotated which will accelerate the process of identifying the QTLs governing the traits of interest in mungbean. A high-resolution LD pattern was identified and several studies in various legumes have been conducted using GBS and WGRS to find the LD pattern and dissect the traits (Zhou et al., 2015; Varshney et al., 2019). This study demonstrated a higher LD estimate (r2 = 0.5) and less extensive LD decay (500–1,000 kb). Similarly, Reddy et al. (2020) also reported a higher LD estimate (r2 = 0.62) and less extensive LD decay (50–100 kb) in mungbean. Comparable LD decay was found (100 kb range) in cultivated genotypes which is slightly higher than 60 kb LD decay as reported for the wild mungbean genotypes (Noble et al., 2018). Liu et al. (2016) reported 200–250 kb as the range for LD decay in soybean; while Hyten et al. (2007) reported 90 to 574 kb LD decay. The structure analysis revealed two population groups, which corresponded with that of the PCA and phylogenetic analysis. Furthermore, this is the first high-resolution determination of LD decay in mungbean with such a large number of genome-wide markers (31,953 SNPs).

Candidate gene identification and validation

A large number of high-quality SNPs (31,953 SNPs) identified from the association mapping panel were used for the mapping of traits like earliness, MYMIV resistance, etc. MLM and BLINK models revealed 338 significant SNPs about the candidate genes involved in the regulation of the studied traits. Orthologous of Arabidopsis thaliana were used to discover the candidate genes involved primarily in the regulation of earliness (flowering time), YMD resistance, and other traits. A total of 44 SNPs shared by both models were found associated with 33 candidate genes have been identified. This study identified NBS-LRR-encoding genes (chr. 9) as having a role in YMD resistance. Similarly, Purwar et al. (2023) also reported the role of NBS-LRR-encoding genes in the imposition of YMD resistance in mungbean and two wild non-progenitors. The TIR-NBS-LRR class of disease resistance protein family (chr. 4 and 9) was found to regulate YMD resistance in mungbean as also reported by Purwar et al. (2023). Depuydt & Vandepoele (2021) also reported the involvement of this gene in a variety of developmental processes and molecular responses, including defense responses.

Similarly, in Vigna mungo, Maiti, Paul & Pal (2012) also reported TIR-NBS-LRR encoding candidate gene regulating MYMIV-resistance. These proteins are encoded by a class of defense-related R genes. Pathogens are recognized by nucleotide-binding site—leucine-rich repeats (NBS-LRRs) proteins, which trigger downstream signaling pathways that activate the defense response. Interestingly, methyl esterification of pectin regulates plant-pathogen interactions thereby regulating YMD resistance. The degree and pattern of methyl esterification alter cell wall characteristics which affects both plant physiological functions and resistance (Lionetti, Cervone & Bellincampi, 2012). In addition, a 30S ribosomal protein S31 in chloroplastic silenced plants showed a delay in expression in the nonhost pathogen (Nagaraj et al., 2016). Similarly, Wang et al. (2017) also observed an interaction between 30S ribosomal subunit protein S11 and Cucumber mosaic virus LS2b protein affecting viral replication, infection, and gene silencing suppressor activity.

For flowering time regulation, several candidate genes (EARLY FLOWERING3, FPA, E3 ubiquitin ligase, CPK32) located on different mungbean chromosomes have been identified. For flowering, an EARLY FLOWERING3 (AT3G21320) gene (chr. 5) was identified as having a role in the regulation of flowering time by encoding a novel protein in Arabidopsis (Hicks, Albertson & Wagner, 2001). The same gene was also reported in barley regulating flowering time by mediating gibberellin production and FLOWERING LOCUS T expression (Boden et al., 2014). Likewise, FPA (AT2G43410) gene (chr. 10) regulates the flowering time via a day-length independent pathway through protein-containing RNA-binding domains in plants (Yu et al., 2021; Macknight et al., 1997). Similarly, Yu et al. (2021) reported that the BDR proteins interact with FPA and the complex is crucial for repressing the flowering time gene FLOWERING LOCUS C (FLC) in Arabidopsis.

In addition, a flower initiation gene E3 ubiquitin-protein ligase DRIP2 gene (chr. 3) was identified which functions by increasing the transcription of WRKY71 gene and thus promoting the expression of Flowering Locus T (FT) and LEAFY (LFY) genes and ultimately counteracting the DELLA gene inhibition (Wang et al., 2022). In addition, several studies revealed the role of ubiquitin E3 ligases in the regulation of flowering time (Pavicic et al., 2017; Lazaro et al., 2012; Peng et al., 2013; Xia et al., 2013). Similarly, Yu et al. (2021) reported that the flowering time control protein FPA of negative transcription elongation factor regulates flowering time in Arabidopsis.

Ca-dependent protein kinase (CPK32) gene (chr. 7) regulates the flowering time via Ca-signaling (Li et al., 2022); while the K-transporter 6 gene (Negishi et al., 2018) and homeobox-leucine zipper protein (HAT22) also regulates the flowering time (Schena & Davis, 1994). These kinases likely contribute to the crossroads of HAT22/ABA and CK signaling during drought response, aiding in leaf senescence (Depuydt & Vandepoele, 2021). The details of different candidate genes and their functions are enumerated in Table 5.

When identified genes were compared with the Vigna radiata genome, some candidate genes were found regulating various other traits. WRKY family transcription factor (WRKY15) mainly regulates the leaf area (15–25%) and plant biomass (Bakshi & Oelmüller, 2014). The WRKY54 and WRKY70 genes were reportedly having negative regulation of leaf senescence in Arabidopsis thaliana (Besseau, Li & Palva, 2012). While, FAR1-RELATED SEQUENCE 4 protein gene regulates the light-mediated signal transduction, chloroplast division, and chlorophyll biosynthesis by regulating the cellular activities through transcriptional activation or suppression of numerous target genes (Ma & Li, 2018). The E3 ubiquitin-protein ligase RIE1 gene has a role in the regulation of leaf senescence. The gene fusion test of the RIE1 promoter with the β-glucuronidase (GUS) gene indicated that this is expressed in several plant tissues (Xu & Li, 2003). Likewise, LOB domain-containing protein 21 gene was found associated with the development of glandular trichomes in plants (Chen et al., 2020).

The role of various transcription factors, including MYB, NAC, GATA, LBD, and ZF on Populus CesA gene expression showed varied expression for the leaf trichomes (Takata & Taniguchi, 2015). Interestingly in cotton, CesA4 was expressed in trichomes, while in Arabidopsis, CesA4, CesA7, and CesA8 were not expressed in the trichomes (Wu, Hu & Liu, 2009; Betancur et al., 2010; Kim et al., 2011).

Thus, breeding for the varieties carrying the genes governing MYMIV resistance will enhance the overall yield. Furthermore, the candidate genes governing flowering time, chlorophyll content, leaf area, and trichome density, provide opportunities for targeted crop improvement. For example, flowering time regulation is crucial for adapting the mungbean crop to the changing climatic conditions. The early-maturing varieties can be developed by manipulating these genes, enabling the farmers to synchronize mungbean cultivation under seasonal variations. Additionally, the genes associated with chlorophyll content, leaf area, and trichome density can be leveraged to enhance photosynthetic efficiency and stress tolerance, contributing to improved crop performance.

Conclusions

The GWAS using 132 diverse mungbean genotypes and 31,953 SNPs for MYMIV resistance, earliness, and various other traits could identify 33 candidate genes regulating these traits. Various common SNPs are identified through MLM and BLINK for traits like flowering time (149), YMD (59), leaf area (74), SPAD value (31), and trichome density (25). The SNPs and candidate genes associated with the earliness and YMD resistance will be of great importance in the mungbean breeding program to incorporate earliness and MYMIV resistance in the desired genotypes. The results demonstrated the efficacy and reliability of implementing high-resolution association mapping in mungbean, which has a narrow genetic base and also very poor availability of diverse genomic resources. However, the candidate genes still need detailed validation using different trait-specific biparental mapping populations and also through a transgenic approach to find their precise role in the regulation of earliness and MYMIV resistance in mungbean. Thus, the genomic resources developed in this study will enrich the mungbean genomics and can be utilized to advance genomic-assisted breeding in mungbean.

Supplemental Information

Supplemental Information 1 The details of Q-matrix

Click here for additional data file.

Supplemental Information 2 The details of SNPs, chromosomes, gene ID and candidate genes

Click here for additional data file.

Supplemental Information 3 Supplementary figures

Click here for additional data file.

Supplemental Information 4 Raw data for Fig. 2, Fig. 3 and Table 2

Click here for additional data file.

The technical support received by Mr. Dilip Kumar (Indian Agricultural Research Institute, New Delhi) is duly acknowledged.

Additional Information and Declarations

Competing Interests

Author Contributions

DNA Deposition

Data Availability

The authors declare there are no competing interests.

Manju Kohli performed the experiments, analyzed the data, prepared figures and/or tables, authored or reviewed drafts of the article, and approved the final draft.

Hina Bansal conceived and designed the experiments, authored or reviewed drafts of the article, and approved the final draft.

Gyan Prakash Mishra conceived and designed the experiments, authored or reviewed drafts of the article, and approved the final draft.

Harsh Kumar Dikshit conceived and designed the experiments, authored or reviewed drafts of the article, and approved the final draft.

Shashidhar B. Reddappa analyzed the data, prepared figures and/or tables, and approved the final draft.

Anirban Roy performed the experiments, prepared figures and/or tables, authored or reviewed drafts of the article, and approved the final draft.

Subodh Kumar Sinha performed the experiments, prepared figures and/or tables, and approved the final draft.

K.M. Shivaprasad analyzed the data, prepared figures and/or tables, and approved the final draft.

Nikki Kumari analyzed the data, prepared figures and/or tables, and approved the final draft.

Atul Kumar performed the experiments, prepared figures and/or tables, and approved the final draft.

Ranjeet R. Kumar performed the experiments, prepared figures and/or tables, and approved the final draft.

Ramakrishnan Madhavan Nair conceived and designed the experiments, prepared figures and/or tables, authored or reviewed drafts of the article, and approved the final draft.

Muraleedhar Aski conceived and designed the experiments, prepared figures and/or tables, authored or reviewed drafts of the article, and approved the final draft.

The following information was supplied regarding the deposition of DNA sequences:

The data is available at NCBI: PRJNA609409.

The following information was supplied regarding data availability:

The raw data is available in the Supplemental File.

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
