# Peer review of "Genome-wide association studies for earliness, MYMIV resistance, and other associated traits in mungbean (Vigna radiata L. Wilczek) using genotyping by sequencing approach"

_PeerJ, doi:10.7717/peerj.16653_

## Round 0.1 · original submission · Major Revisions

Based on the reviewer's reports, the paper is being send back to the authors for MAJOR REVISION. Please address all the queries raised by the reviewers and resubmit point by point response.

**Language Note:** PeerJ staff have identified that the English language needs to be improved. When you prepare your next revision, please either (i) have a colleague who is proficient in English and familiar with the subject matter review your manuscript, or (ii) contact a professional editing service to review your manuscript. PeerJ can provide language editing services - you can contact us at [email protected] for pricing (be sure to provide your manuscript number and title). – PeerJ Staff

Reviewer 1 ·

Basic reporting

In this manuscript, authors report GWAS of MYMIV resistance in a set of mungbean germplasm.
MYMV is a serious and the most important disease of the mungbean production in South and Southeast Asia. Finding in this study may be useful for breeding of the mungbean. However, the manuscript contains big flaws regarding experimental design and data collection.

Experimental design

1. In this study, authors used augmented design for evaluating MYMIV disease resistance in the GWAS germplasm set. However, the experiment is done only one season. It is generally accepted that QTL analysis, both by biparental mapping and GWAS, phenotyping should be done in at least 2 environment.
In addition, yellow mosaic disease (YMD) in the mungbean can be cuased by the virus MYMV and MYMIV. The symthoms of the YMD caused by these two virus are very similar. Since the authors did not identify the causal agent of the YMD disease. How can you be sure that the phenotypes you measure are accurate.
2. How many plants in each germplasm were used to measure/phenotype for the disease. And in each plant how many leaves and what position of the leaves were used?

Validity of the findings

It seems that candidate genes identified for the traits reported in this study are different from other previous reported in the mungbean. The author did not validate the finding by biparental QTL mapping.
In addition, the authors did not mention any previous reports in the mungbean, especially the MYD. I suggest that authors added more information regarding QTL controlling MYD in the mungbean in the Introduction section.

Additional comments

1. What was the mungbean reference genome used for GWAS and candidate gene identification used in this study? I think you used the reference genome reported by Kang et al. (2014).
2. It would be useful to readers if authors include a table showing sequence and position of all SNPs associting to each trait.

·

Basic reporting

The article seems to be well written in English.
Introduction section - Information regarding the genetics of mungbean is very important in association studies. In this manuscript, it is partially provided. It seems to be spreading around the introduction section. Authors can improve by linking them together capturing important areas such as ploidy levels, chromosome number and current reference genome availability. There have been a growing number of publications in mungbean GWAS in recent time that the authors can cite from.

Experimental design

The study design is ok.
Materials & Methods – Line 40: “… population structure was determined using STRUCTURE v2.3.4 (ref????)..” I think the authors have quoted the wrong reference here!
Also, include the simulation criteria for structure analysis in M&M section. Not in your discussion! I will suggest the structure criteria (lines 184-185) be removed at the Discussion and placed at the M&M section.
Again, there seems to be a confusion with the symbols of Delta K. In various part of the manuscript the symbols seems to be mis represented (e.g. Line 23).

Validity of the findings

Did the authors use K = 2 sub-population as their co-variates in association mapping analysis? If yes, they should include the Q-matrix in the supplementary file to validate the reproducibility of the results, so the results presented is worthy.
Results – At the result section (lines 298-299), there could be an ambiguity of presented result. The authors should clearly verify why subpopulation of 2 (K=2) can coincide with the PCA analysis and the construction of phylogenetic tree. In association studies, sometimes the determination population structure can influence the PCA analysis and phylogenetic tree. Hence, causing a premature determination of the PCAs. Please do clarify on how population structure does agree with PCA and phylogenetic tree?

Additional comments

Line 38-39 – Sentence needs to be re-written; especially at line 39. “ 7million hectares are…”. Suggest to insert a connective word or remove the word “area”.
Line 141 – Please reference your “R” Environment.
Line 156 – “…measured by ‘R software’….” Are the authors sure it is ‘R software’? I would think otherwise, that ‘R’ is an online environment. Please restate that in other parts of the manuscript as well.
Line 700 - Delta “L” plot. Are the authors sure it is Delta “L” or Delta K? Please verify??

Reviewer 3 ·

Basic reporting

The study holds significant importance for the agricultural and genetic research community. Mungbean is a vital crop in many tropical and subtropical regions, providing a cheap source of protein in developing countries. However, it faces challenges such as yellow mosaic disease (YMD) and the need for early-maturing varieties to adapt to changing climatic conditions.

This study employed advanced genotyping techniques to analyze a diverse panel of mungbean genotypes for traits like flowering time, YMD resistance, chlorophyll content (SPAD value), leaf area, and trichome density. By identifying 31,953 high-quality SNPs across the mungbean genome, the researchers conducted genome-wide association studies (GWAS) to link these genetic markers with specific traits. Notable candidate genes associated with flowering time, YMD resistance, chlorophyll content, leaf area, and trichomes were identified. In-silico validation of these candidate genes through digital gene expression analysis further supported their roles in regulating these traits.

The findings from this study provide valuable insights for marker-assisted breeding programs aimed at developing mungbean varieties that are resistant to YMD, early-maturing, and possess desirable agronomic traits. This research contributes to improving mungbean crop productivity, addressing disease challenges, and enhancing food security, particularly in regions where mungbean is a dietary staple. Overall, this study bridges the gap between genomics and crop improvement, offering practical solutions for sustainable agriculture in mungbean production. I have some suggestions to improve this study as given below:
a. Please clarify the term "AUDPC" for readers who may not be familiar with it.
b. The introduction provides a comprehensive overview of mungbean, YMD, and its significance, but it would be helpful to briefly introduce GWAS and its relevance to the study.
c. Ensure consistency in abbreviations and their explanations throughout the paper. For example, MYMV and MYMIV are mentioned separately, but their full names are not provided.
d. The methods section should include more details on the genotyping by sequencing (GBS) process, such as library preparation, sequencing platforms used, and quality control steps.

Experimental design

a. Please elaborate on the rationale behind selecting the specific traits (flowering time, YMD resistance, SPAD value, trichome density, and leaf area) for investigation. Were these traits chosen based on prior knowledge or hypotheses?
b. In the phenotypic observations section, provide more details on the methodology used for recording trichome density and leaf area. Were these measurements taken from multiple individuals and averaged for each genotype?
c. How were the 132 mungbean genotypes selected for the AM panel? Were there specific criteria for inclusion, and were they representative of the genetic diversity within the species?
d. Please provide more information on the statistical methods used to calculate broad sense heritability (h2) for various traits. Were these estimates based on a single season or multiple seasons of data?

Validity of the findings

a. In the results section, there is mention of significant associations between SNPs and traits, but the significance thresholds and criteria for declaring significance should be explicitly stated. How was the Bonferroni correction threshold determined?
b. For the candidate genes identified, please discuss their known functions in Arabidopsis or other relevant species. How do these functions relate to the observed trait associations in mungbean?
c. In the digital gene expression analysis, it would be helpful to provide more context on how the expression of identified candidate genes correlates with the phenotypic variation observed in the study.
d. Were any potential limitations or challenges encountered during the GBS and GWAS process? Were there any population substructures or potential confounding factors that might have affected the results?
e. In the discussion section, please discuss the practical implications of the findings for mungbean breeding and agriculture. How might the identified QTLs and candidate genes be used for crop improvement?

---

## Round 0.2 · accepted · Accept

Based on the reviewer's evaluations, the paper has significantly improved and is acceptable for publication.

Reviewer 3 ·

Basic reporting

The manuscript demonstrates a commendable standard of basic reporting, meeting the criteria for acceptance

Experimental design

The experimental design of the study is robust and well-executed

Validity of the findings

The validity of the findings in this manuscript is a critical aspect that has been meticulously addressed, supporting the case for acceptance. The methods employed in data collection and analysis are sound, aligning with the research questions and objectives.

Additional comments

The manuscript is significantly improved in context of suggestions. The authors have responded to all the queries. I think the manuscript is now ready for publication.